# A Universal and Quantitative PCR Strategy for Detection and Epidemiologic Analysis of Canine Papillomavirus (CPV)

**DOI:** 10.3390/ijms26094391

**Published:** 2025-05-06

**Authors:** Dan Zhou, Kaixin Wang, Youming Yuan, Yalan Li, Richard Schlegel, Aibing Wang, Hang Yuan

**Affiliations:** 1Department of Pathology, Georgetown University Medical School, Washington, DC 20057, USA; dz32@georgetown.edu (D.Z.);; 2College of Veterinary Medicine, Hunan Agricultural University, Changsha 410128, China; wangkaixin0708@163.com (K.W.); yalanli@stu.hunau.edu.cn (Y.L.); 3College of Food Science & Nutritional Engineering, China Agricultural University, Beijing 100083, China; yuanyouming06@outlook.com

**Keywords:** canine papillomavirus (CPV), polymerase chain reaction (PCR), CPV epidemiology, viral genotyping, diagnostic techniques

## Abstract

Canine papillomavirus (CPV) infection leads to a range of clinical manifestations from benign warts to malignant tumors in dogs, posing significant challenges in veterinary medicine due to its diverse genotypic spectrum. This study introduced broad-range and robust polymerase chain reaction (PCR) assays designed to enhance the detection, identification, and quantification of multiple CPV genotypes. By using both universal and genotype-specific primers, this protocol significantly improved diagnostic specificity and sensitivity across the 23 known CPV genotypes compared to previously described ones. The primers were designed to target conserved regions across all genotypes for general detection, as well as specific regions in the predominant genotypes CPV1 and CPV2 for detailed analysis. Validation of this protocol using synthetic plasmids and clinical samples confirmed its enhanced performance over traditional methods, as demonstrated by higher specificity and sensitivity. Additionally, the application of this PCR approach in a series of epidemiological studies provided novel insights into the distribution and prevalence of CPV genotypes, highlighting its potential utility in shaping targeted vaccination and clinical management strategies. Furthermore, the quantitative capability of this established protocol allowed for monitoring viral loads in clinical cases, offering a valuable tool for assessing treatment efficacy and disease progression. Further validation through larger-scale clinical studies will be crucial to substantiate the diagnostic accuracy and epidemiological value of the assays.

## 1. Introduction

Papillomaviruses (PVs) are host species-specific, circular, double-stranded DNA viruses that primarily induce skin and mucous membrane disorders in a wide range of vertebrate species, including humans, birds, and reptiles [1,2,3]. Canine papillomaviruses (CPVs) are the viral agents responsible for a wide spectrum of cutaneous and mucosal diseases in domestic dogs. These pathological conditions range from benign warts and oral papillomatosis to cutaneous papillomas, viral pigmented plaques, and severe malignancies like squamous cell carcinoma, thereby posing significant implications for animal health and welfare [2]. To date, over 23 genotypes of CPVs have been identified and classified based on their L1 sequences [2,4,5,6].

Currently, several techniques have been adopted to detect the viral etiology of canine papillomatosis. These include polymerase chain reaction (PCR) [6,7]; rolling circle amplification (RCA) [8]; DNA in situ hybridization (ISH) for detecting CPV genomic DNA [9]; histopathology of hematoxylin and eosin (H&E)-stained tissue for revealing neoplastic tissue growth and cytopathic effects associated with papillomavirus infection, while not being specific to CPV [10]; immunohistochemistry (IHC) for probing L1 viral antigen [11]; and transmission electron microscopy (TEM) for visualizing viral particles within pathological lesions [12]. Additionally, the application of next-generation sequencing technology (NGS) for the detection of CPV has also been described [13]. Notably, each of these approaches has its strengths and shortcomings. For instance, the high cost of NGS prevents it from becoming a routine method for CPV detection [13]. Therefore, the development of accurate, convenient, cost-effective methods for detecting and genotyping of CPV infections is crucial for effective clinical management and epidemiological studies.

The genetic diversity of CPVs, which includes almost two dozen genotypes, underscores the need for the development of a universal (general or broad-range) detection method [2,3]. Notably, CPV types 1 (CPV1) and 2 (CPV2) are frequently identified as the predominant strains in papillomas samples, often including a certain proportion of co-infections [14]. Despite this prevalence, no PCR method has been reported that can simultaneously detect these major two types. Furthermore, there may be a potential relationship between the viral load of CPV1 and CPV2 and their clinical manifestations, but an effective quantitative approach for assessing viral copies has yet to be developed. Given that PCR is regarded as the gold standard for the detection of CPV infections due to its high sensitivity and specificity [15], we aimed to develop a PCR-based strategy to address these diagnostic limitations.

In this manuscript, we detailed the design of a novel PCR-based assay and its application to the detection of CPVs in both previously identified and newly collected clinical samples, confirming its convenience, sensitivity, and specificity. We outlined the methodology for primer design, targeting conserved regions across 23 CPV genotypes for universal detection, and specific regions in predominant genotypes CPV1 and CPV2 for detailed analysis. Additionally, we presented findings from partial epidemiological investigations utilizing this established approach, highlighting its efficacy in identifying and quantifying CPVs. This protocol filled in a critical gap in CPV diagnostics, offering a robust tool for clinical management and epidemiological studies.

## 2. Results

### 2.1. Analyzing CPV L1 Sequences to Design PCR Primer Sets

Following multiple sequence alignments, two relatively conserved regions within the CPV L1 genes were identified and subsequently used to design a universal primer pair (Figure 1A). Several nucleotides in the upstream and downstream primer sequences were substituted with degenerate bases, following the guidelines established by the International Union of Biochemistry (IUB). To develop a PCR assay capable of distinguishing CPV1 and CPV2, a sequence BLAST was performed. A completely conserved region in the L1 sequence, shared by both CPV1 and CPV2, was identified and employed as the common upstream primer for the assay (Figure 1B). Multiple downstream primers were then specifically designed to target the L1 regions of either CPV1 L1 or CPV2 (Figure 1C,D), resulting in PCR amplicons with a size difference of 50 or 100 base pairs (bp) in length. Additionally, two primer pairs were designed to amplify the full-length L1 genes of CPV1 and CPV2 separately. The primer names, combinations, amplicon sizes, and their respective usages are detailed in Table 1 and illustrated in Figure 1.

### 2.2. Establishing a Universal PCR Detection Assay for CPVs

To develop a universal PCR assay capable of detecting all 23 identified CPV genotypes, a general primer pair (CPV-General23-F0+CPV-General23-R0) was designed to detect the presence of CPV nucleic acids. Using this primer pair and plasmid DNA controls from each of the twenty-three types of CPV as templates, a specific DNA product approximately 700 bp in length was amplified from all twenty-three CPV genotypes, while no bands were observed in the absence of CPV DNA (Figure 2A). This newly designed universal primer pair was then compared to a previously described primer set (namely, canPVf/FAP64, referred to here as CPV-General7-F+CPV-General7-R), which had been reported to detect CPV genotypes 1-7 [16]. This comparison focused on the assay’s sensitivity and ability to detect CPV nucleic acids using whole DNAs extracted from 16 clinical samples containing CPV genotypes 1-7. The results demonstrated that although DNA concentration was not quantified PCR reactions using our universal 23 primer pair consistently yielded stronger and specific bands across all 16 samples. In contrast, the reactions with the CPV-General7-F+CPV-General7-R primer pair produced weak DNA signals and failed to detect one sample (Figure 2B). Notably, no cross-reaction with HPV-L1 DNA was observed when using our universal primer set, indicating its better specificity and sensitivity based on comparative testing.

To further confirm the sensitivity of the assay with general primers, a semi-quantitative comparative PCR assay was performed on serially diluted CPV1 L1 plasmid samples. As illustrated in Figure 2C, the PCR assay using our general primer set could detect CPV at a sensitivity of as low as 10^3^ copies per microliter of DNA. In contrast, the assay using CPV-General7-F+CPV-General7-R primer set detected CPV at a sensitivity of 10^5^ copies per microliter of DNA. Therefore, these results prove that the universal primer designed in this study exhibits improved specificity and sensitivity compared to the primer set reported earlier [16].

### 2.3. Developing a PCR Assay for Detection and Simultaneous Differentiation of CPV1 and CPV2

Previously, genotyping canine parvoviruses (CPVs) using PCR required a series of type-specific primers, often involving multiple PCR rounds or the simultaneous use of all primers, making the process both time-consuming and labor-intensive [2]. Given the high prevalence of CPV1 and CPV2 infections in clinical settings [2], a PCR assay that can simultaneously differentiate between these two genotypes is highly valuable. To address this need, we designed two sets of three-primer combinations: CPV1/2 L1-F + CPV1 L1-R1 + CPV2 L1-R1 and CPV1/2 L1-F + CPV1 L1-R2 + CPV2 L1-R2. These combinations were intended to detect the presence of CPV1 and CPV2 nucleic acids, either individually or simultaneously. Initial tests were conducted using single and combined plasmids containing CPV1 L1 and/or CPV2 L1 genes as the templates. Interestingly, both sets of primer combinations consistently yielded specific and expected DNA bands. As shown in Figure 3A, the presence of both CPV1 L1 and CPV2 L1 plasmids generated two distinct bands, one of 400 bp for CPV1 and another of 300 bp for CPV2, while the existence of only one type of plasmid resulted in a single band, either 400 bp or 300 bp, respectively. Similarly, using the second three-primer combination led to the appearance of a 450 bp band for CPV1 and a 350 bp band for CPV2 when these plasmids were present singly or combined. Subsequently, the three-primer combination (CPV1/2L1-F+CPV1L1-R1+CPV2L1-R1) was further utilized to detect individual or simultaneous presence of CPV1 and CPV2 nucleic acids in whole DNAs extracted from 18 clinical canine wart samples (S), either singly or co-infected with CPV1 and/or CPV2 (Figure 3B). Accurate results were consistently and reproducibly achieved across these clinical samples, completely aligning with the previous experimental data obtained using other methods. Notably, several DNA bands corresponding to CPV1 was weaker than those for CPV2, possibly due to varying DNA concentrations within the respective samples. Despite this, the results confirmed that both sets of three-primer combinations are sensitive and specific for detecting single and co-existing CPV1 and CPV2 infections. Taken together, this newly established assay streamlines and simplifies the detection of both single and mixed infections of CPV1 and CPV2, making the process more straightforward and convenient.

### 2.4. Implementing a Real-Time PCR Technique to Quantify the Viral Content of CPV1 and CPV2

Quantitative analysis of the relationship between virus copy number and disease outcome may be critical for understanding papillomavirus pathogenesis. To facilitate the analysis of viral content in clinical samples, we established a real-time quantitative PCR (RT-qPCR) assay specifically targeting the most prevalent CPV1 and CPV2 genotypes. Standard curves were generated using respective 10-fold diluted plasmid controls, ranging from 109 to 102 copies/µL, to calculate the viral DNA content. Melt curve analysis indicated the absence of primer–dimer artifacts and verified the specificity of the reaction, as illustrated in Figure 4A,B. Importantly, the standard curves demonstrated a robust linear relationship between the initial template concentrations and Ct values, as evidenced by an excellent correlation coefficient (R2), the slope of the equation and the amplification efficiency (E). These parameters for CPV1 were 0.9986, −3.419, and 96.10%, respectively, while they were 0.9951, −3.505, and 92.89% for CPV2 (Figure 4C,D). Using the RT-qPCR assay, the viral contents of four CPV1 and two CPV2 clinical samples were quantified, as indicated in Table 2. Additionally, gel electrophoresis for RT-qPCR reaction products from these clinic samples revealed the CPV1/2-L1-F0+CPV1-L1-R0 primer pair yielded a specific 190 bp product for CPV1, while the CPV1/2-L1-F0+CPV2-L1-R0 pair generated a unique 140 bp product for CPV2 (Figure 4E). These data validate the sensitivity and specificity of this newly established RT-qPCR method for determining CPV1 and CPV2 viral content in clinical settings.

### 2.5. Investigating the Epidemiology of CPVs

Finally, we analyzed the epidemiology of CPVs in CPV-infected wart samples collected from various animal hospitals across the USA. These samples were initially identified by other methods (multiple-round PCR and RCA with restrictive enzyme digestion) and subsequently validated using the newly established method. Additionally, we examined clinical samples collected from animal hospitals, rescue centers, and breeding facilities throughout China, also using this newly established method.

The data presented in Table 3 provide a detailed CPV epidemiological analysis across the USA over a span of five years, from 2017 to 2021. The table organizes the data by year and categorizes them based on the presence of CPV genotypes, including CPV1, CPV2, other CPV genotypes, and co-infections of CPV1 and CPV2. Throughout the observed period, several features of CPV epidemiology were noted: (A) CPV1 emerged as the most commonly detected genotype, appearing in 109 out of the total 142 samples, occupying 79.31% of all cases, highlighting CPV1’s dominance in clinical presentations. (B) CPV2, less common relative to CPV1, was detected in 11 samples, representing 7.75% of the total cases, while its co-infection with CPV1 was documented in 10 samples, accounting for 7.04% of total cases, bringing CPV2’s overall involvement to nearly 15% of all cases. (C) Other CPV genotypes were identified in 12 samples, making up a small fraction of 8.45%, suggesting their occasional emergence. (D) Annual data revealed variability in genotype prevalence and detection rates; however, CPV1 consistently maintained high detection rates each year. This extensive dataset provides valuable insights into the epidemiological trends of CPV infections, aiding in the understanding of its spread and prevalence within the canine population.

An analysis of CPV epidemiology was conducted on oral swab samples collected from different animal facilities across China in 2019, including animal hospitals, rescue centers, and breeding facilities. Whole DNAs extracted from these samples were subjected to quality analysis, universal detection and sequencing, and genotyping through sequential PCR with the dogGAPDH primer set, the universal primer pair, and the CPV1/CPV2 three-primer set. The data were primarily categorized based on the identification of CPV1 and CPV2, and instances of co-infections involving both genotypes (Table 4). Results from animal hospitals revealed the absence of single CPV1 infections, with CPV2 detected in 4 out of 21 samples (19.05%), and co-infections observed in 2 samples (9.52%), suggesting a moderate prevalence of CPV2 and co-infections, but no single occurrence of CPV1 in this setting. Conversely, data from animal rescue centers showed a significantly higher incidence of CPVs, with CPV1 found in 32 out of 51 samples (62.74%), CPV2 seen in 14 out of 51 cases (27.45%), and their co-infections present in 5 samples (9.80%), indicating a high prevalence of CPV in this environment. At the breeding facilities, CPV1 was detected in 6 out of 18 samples (33.33%), while CPV2 appeared more frequently in 10 samples (55.56%), and co-infections were noted in 2 samples (11.11%), similarly pointing to a higher prevalence of these two major CPV genotypes in this professional setting. Additionally, no other CPV genotypes were found, as verified by sequencing the general primer set PCR products. Overall, the results highlight notable variations in CPV genotype prevalence across different types of animal care facilities, reflecting differing risks and transmission dynamics associated with each environment.

## 3. Materials and Methods

A schematic diagram illustrating the experimental procedure and content of this study is demonstrated in Figure 5.

### 3.1. Sequence Analysis and Primer Design

The genomic sequences of 23 CPV genotypes were retrieved from the NCBI database, with their corresponding GenBank accession numbers listed in Appendix A. Subsequently, the positions and sequences of the L1 genes for each of the 23 CPV genotypes were individually determined based on their annotated genomic locations and aligned using the Multalin tool, accessible at multalin.toulouse.inra.fr/multalin/ (accessed on 19 March 2025). Additionally, the potential consensus sequences between the L1 of CPV1 and CPV2 were examined by BLAST (version 1.4.0). Primers targeting the L1 gene sequences of CPVs were designed to ensure broad-range detection. A general or universal or broad-range primer pair, hereafter referred to as the same concept, was developed to detect all 23 CPV genotypes, while type-specific primers, targeting the most prevalent types CPV1 and CPV2, were designed with a common upstream primer and distinct downstream primers to produce amplicons of variable sizes, facilitating differentiation between these two types. Primer pairs were also developed to amplify the full-length L1 gene of CPV1 and CPV2. All primers were computationally analyzed and synthesized by Genscript (Nanjing, China) for validation. Additionally, previously reported primer sets, including canPVf/FAP64 and dogGAPDHf/dogGAPDHr primer pairs, were also included for comparison and genomic DNA quality analysis, respectively [1].

### 3.2. Preparation of Standard or Positive Plasmids

Full-length L1 genes of CPV1 and CPV2 were amplified via PCR using the corresponding primers listed in Table 1, with genomic DNA extracted from previously isolated clinical samples, which had been identified using other methods, serving as the templates. The amplified DNA fragments underwent DNA gel separation, excision, and purification following the manufacturer’s instructions (Qiagen, Hilden, Germany). These purified fragments were then cloned into the pEF-GFP vector to substitute the excised GFP segment. Both plasmids were prepared with a minipreparation kit (Qiagen, Hilden, Germany), and their DNA concentrations were determined by a Nanodrop spectrophotometer (Thermo Fisher Scientific, Waltham, MA, USA). To serve as standard control samples, these two plasmids were serially diluted 10-fold and converted into copy numbers according to the formula: number of copies (molecules) = X ng × 6.022 × 10^23^ molecules/mole/N × 660 ng/mole × 10^9^ ng/g, where X is the concentration in ng, N is DNA length, 660 g/mol represents the average mass of 1 bp dsDNA, 6.022 × 10^23^ is Avogadro’s constant, and 1 × 10^9^ is the conversion factor.

Meanwhile, partial L1 gene sequences of CPV genotypes 3-23, with detail provided in Appendix A, were synthesized and individually inserted into pUC57 vector. These constructed plasmids were then diluted to a concentration of 10 ng/µL.

### 3.3. Sample Collection and DNA Preparation

CPV-infected wart samples were collected between 2017 and 2021 from animal hospitals across the United States. Concurrently, clinical samples were obtained in 2019 by swabbing the oral cavities of dogs at animal hospitals, rescue centers, or breeding facilities throughout China, as previously described [16], in accordance with ethical guidelines. All animal experiments received approval from the Institution Animal Care and Use Committee (IACUC) of the respective universities. Total canine DNA was isolated from these specimens using the DNeasy Blood & Tissue Kit (Qiagen, Hilden, Germany) in accordance with the manufacturer’s instructions. The wart samples from the USA were identified by type-specific PCR or rolling circle amplification (RCA) followed by restrictive enzyme digestion [8] and are hereafter referred to as previously identified samples.

### 3.4. PCR and Real-Time Quantitative PCR (RT-qPCR)

Conventional PCR was performed on a T100 thermal cycler (BioRad, Hercules, CA, USA) and optimized for maximum efficiency and specificity. For amplification with the universal primer pair, the PCR conditions were set as follows: denaturation at 95 °C for 2 min, followed by 35 cycles of 30 s at 95 °C for denaturation, 30 s at 50 °C for annealing, and 45 s at 72 °C for extension, with a final extension at 72 °C for 10 min. The reaction mixture comprised 25 µL 2 × PCR Master mix (Thermo Fisher Scientific, USA), 2 µL (20 µM) primer pair, 1 µL DNA template, H2O to a total volume of 50 µL. When employing the three primer pairs to differentiate between single or co-infections of CPV1 and CPV2, the protocol was set as follows: initial denaturation at 95 °C for 2 min, followed by 35 cycles of 30 s at 95 °C, 30 s at 60 °C and 30 s at 72 °C, with a final extension at 72 °C for 10 min. The reaction mixture included 10 µL 2× PCR Master mix (Thermo Fisher Scientific, USA), 3 µL (20 µM) primer pair, 1 µL DNA template, H_2_O to a total volume of 20 µL. For the amplification of the full-length L1 genes of CPV1 and CPV2, the PCR conditions were initial denaturation at 95 °C for 2 min, followed by 35 cycles of 30 s at 95 °C, 30 s at 55 °C, and 90 s at 72 °C, with a final extension at 72 °C for 10 min. The reaction mixture contained 10 µL 2× PCR Master mix (Thermo Fisher Scientific, USA), 2 µL of 20 µM primer pair, 1 µL of DNA template, H_2_O to a total volume of 20 µL. Following PCR, the products were analyzed by electrophoresis on a 1.0 or 1.2% agarose gel stained with ethidium bromide. The sizes of the bands were estimated according to 100 bp or 1 kb GeneRuler DNA ladder (Thermo Fisher Scientific, USA) or DL 5000 or DL 2000 DNA marker (Takara Bio, Beijing, China).

To quantify the viral DNA content, a standard curve was established for each experiment by co-amplifying eight consecutive 10-fold serial dilutions of CPV1 and CPV2 standard plasmids. The concentrations of CPV viral DNA in the samples were then defined by plotting the Ct (cycle threshold) values against the standard curve and expressed as copies/µL. RT-qPCR were performed by using iQ SYBR Green Supermix (BioRad, USA) and quantitative primer pairs listed in Table 1 on the CFX96 Real-time System (BioRad, USA). Reactions were conducted in triplicate, and the results were averaged for each sample. The amplification reactions were carried out in a total volume of 20 µL, consisting of 10 µL of iQ SYBR Green Supermix (BioRad, USA), 1 µL forward and reverse primer mixture, 1 µL of template DNA, and 8 µL of H2O. The amplification protocol included a preincubation step at 95 °C for 3 min and followed by 40 cycles of at 95 °C for 10 s and 55 °C for 30 s. Post-amplification, melt curve analysis was performed to confirm the specificity of the target amplification, consisting of an initial step at 95 °C for 10 s, followed by a melt curve 65.0 to 95 °C, incrementing 0.5 °C every 5 s. The quantification cycle (Ct) was determined as the cycle number at which the fluorescence exceeded a threshold level indicative of exponential amplification.

### 3.5. Epidemiological Investigation

Following the validation against standard or positive plasmids and previously identified clinical samples, the primers mentioned previously were applied to detect the presence of CPVs in clinically collected samples. Genomic DNAs prepared from these samples were firstly tested for quality and host specificity by PCR using dog GAPDH primer pair [1]. Subsequently, a second round of PCR was performed with the universal primer pair (CPV-General23-F0/CPV-General23-R0). PCR products displaying the target DNA size were then purified, cloned, and sequenced. For samples that tested positive in the universal PCR, a third round PCR with a three-primer pair was conducted to determine whether they were infected with CPV1 and/or CPV2.

Finally, an epidemiological analysis of CPV infection data from both previously identified and newly collected samples was conducted, with the results organized by year and source.

## 4. Discussion

At the time this study was completed, new CPV genotypes continued to emerge [2,6,17,18], underscoring the dynamic nature of this field. Various diagnostic methods have been employed to detect CPV infections, including PCR, RCA [19], ISH [9], CPV DNA chromogenic in situ hybridization (DNA-CISH) and RNAscope in situ hybridization (RNA-ISH) [4], IHC [20,21], H&E [21], TEM [11,22], and NGS [13]. These diverse genotypes of CPV can cause a range of clinical manifestations from benign warts to malignant tumors in dogs. In particular, several CPV genotypes have been linked to malignant lesions and identified as significant risk factors for the development of squamous cell carcinomas (SCCs) in dogs [19,23,24]. Considering that PCR remains the cornerstone for effective CPV detection, offering advantages such as the capability to perform both qualitative and quantitative analyses, along with convenience and rapidity over other methods [2,4,17,25,26], further enhancements to this method are crucial for broad practical applications in both laboratory and clinical settings. In this context, universal and quantitative PCR assays for CPV detection, as detailed in this study, represent an obvious advancement in the field of veterinary diagnostics. The new protocol not only improves diagnostic efficiency but also deepens our understanding of CPV epidemiology, thereby offering a robust tool for managing and controlling CPV infections across canine populations.

Conventional PCR, while widely used, typically targets specific CPV genotypes, often requiring one pair of primers per CPV genotype [8]. Previously described degenerate or broad-range primer pairs targeting the L1 gene covered less than one-third of identified CPV genotypes [27,28,29,30]. Importantly, the comparison between the previous reported and frequently used primer pair and the one developed in this study clearly demonstrated that the latter offers better sensitivity and specificity. Furthermore, the universal/degenerate primer pair described here covered at least 23 genotypes, significantly expanding its applicability. Additionally, our developed three-primer combinations simplified and enhanced the identification of individual or co-infections of the major genotypes, CPV1 and CPV2 [14].

Besides genotyping, assessing viral load can help explore its correlation with disease progress and severity. In particular, there is a necessity to genotype and quantify CPV1 and CPV2, as they account for a substantial proportion of infections in canine populations. Moreover, a correlation between viral load and disease severity has been noticed [8,14,19,31]. Therefore, incorporating quantitative analysis is also beneficial in clinical settings, where understanding the viral burden can guide treatment decisions, including therapy and prognosis for CPV infections. Previously, we established a probe-based RT-qPCR approach to determine the copy number of CPV1 by targeting its E6 gene [8]. In the current study, we developed a new RT-qPCR method for analyzing the viral content of both CPV1 and CPV2 by targeting the L1 gene. The expansion of this quantification assay, coupled with simplified genotyping for these two major CPV genotypes, enables routine and straightforward monitoring of their prevalence and facilitates their diagnosis.

The application of the aforementioned PCR assays to detect previously analyzed and newly collected samples not only verified their effectiveness but also provided new insights into the epidemiology of CPV. Most CPV-infected samples, collected in the form of warts from different parts of dogs across the USA, were identified as either single or co-infections of CPV1 and CPV2. Notably, their single and co-infections accounted for a large proportion of over 75%, with the highest incidence recorded at 91.66%. These findings not only support previous conclusion that CPV1 and CPV2 are the predominant types [4,14,26] but also highlight the importance and priority of developing vaccines against these two CPV genotypes in the future. The necessity for such vaccines is further supported by the analysis of oral swab samples of dogs from various settings in China, where only CPV1 and CPV2 were identified. The breeding environments appeared to significantly influence the epidemiology of CPVs, as evidenced by the infection rate of CPV1 and CPV2 reaching 100% in animal rescue centers and professional dog breeding facilities. This high prevalence might be attributed to the mixed feeding and housing of the dogs, thereby facilitating frequent contact and transmission of CPV viruses.

In conclusion, the development of universal, convenient, and quantitative PCR assays marks an advancement in the diagnostic tools available for CPV detection. These assays not only address the limitations of previous methods but also enhance our understanding of CPV epidemiology, which is essential for effective disease management and control in canine populations. Notably, further validation through larger-scale clinical samples or studies will be essential to confirm the diagnostic accuracy and epidemiological value of the established strategy. Moreover, future research should also aim to expand the assays’ capabilities to encompass emerging CPV genotypes and explore potential integration with other diagnostic platforms to achieve broader diagnostic coverage and enhanced efficiency.

## Figures and Tables

**Figure 1 ijms-26-04391-f001:**
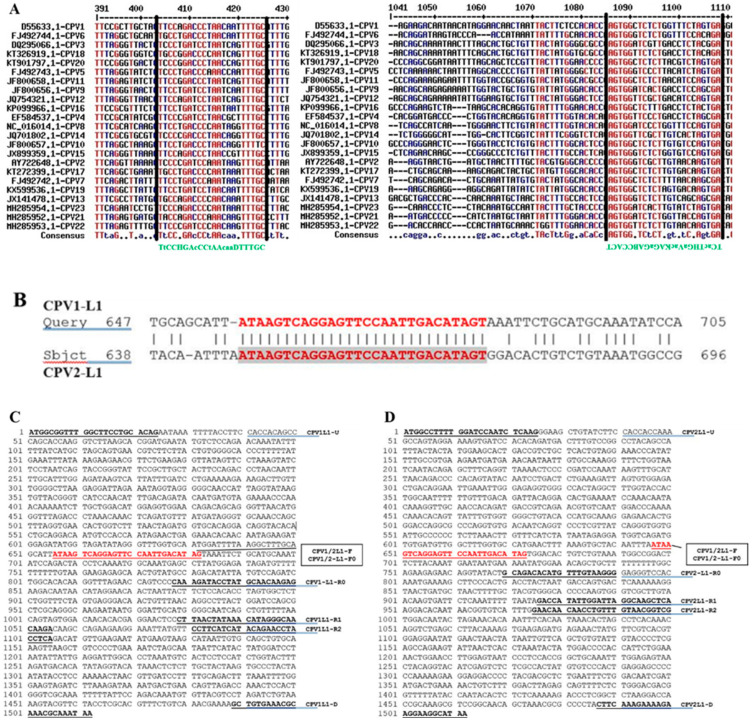
Sequence analysis of CPVs and design of primers. (**A**) Two relative conserved regions (marked between two solid lines) were identified through aligning the L1 sequences of 23 known CPV genotypes, these regions were used to design general/universal/broad-range primers (marked in green). (**B**) A complete consensus sequence between CPV1 L1 and CPV2 L1 was identified and used as the common forward primer for these two genotypes. (**C**) All designed primers for CPV1 are indicated in its L1 sequence. (**D**) All designed primers for CPV2 are indicated in its L1 sequence.

**Figure 2 ijms-26-04391-f002:**
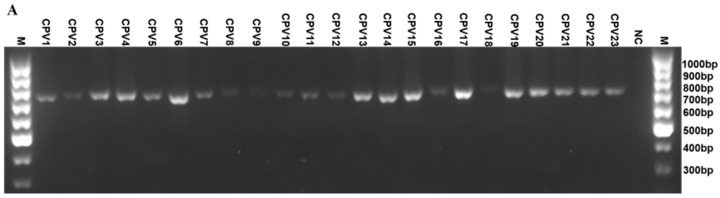
Detection of 23 CPV genotypes using a general primer pair. (**A**) The general primer pair (CPV-General23-F0+CPV-General23-R0) was applied to detect the presence of CPV nucleic acids, employing individual plasmid controls containing either full-length or partial L1 gene DNA fragment as the templates. (**B**) Two sets of general primer pairs (CPV-General23-F0+CPV-General23-R0, upper; CPV-General7-F+CPV-General7-R, bottom) were applied to detect the presence of CPV nucleic acids in whole DNAs extracted from 16 clinical samples harboring CPV1-7 genotypes. (**C**) Comparative PCR assays were performed using the CPV-General23-F0+CPV-General23-R0 primer pair (left, 704 bp) and the CPV-General7-F+CPV-General7-R primer pair cited in the literature (right, 389 bp), on serially diluted CPV1 L1 plasmid samples (1–10: ranging from 6.25 × 10^1−10^. PC: positive control; NC: negative control; M: DNA ladder; HPV: HPV-L1 DNA.

**Figure 3 ijms-26-04391-f003:**
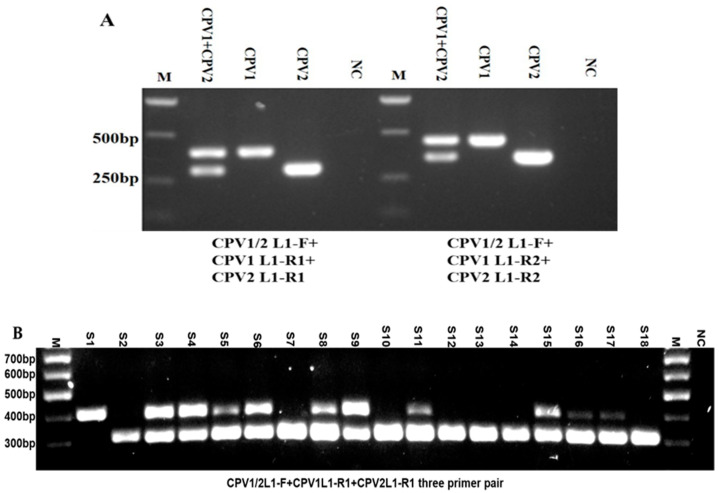
Detection of individual or simultaneous presence of CPV1 and CPV2 using two sets of three-primer pairs. (**A**) Two sets of three-primer pairs CPV1/2 L1-F+CPV1 L1-R1+CPV2 L1-R1 (left) and CPV1/2 L1-F+CPV1 L1-R2+CPV2 L1-R2 (right) were applied to detect the individual or simultaneous presence of CPV1 and CPV2 nucleic acids, using single and combined plasmid controls containing CPV1 L1 and/or CPV2 L1 genes as the templates; (**B**) the primer set CPV1/2 L1-F+CPV1 L1-R1+CPV2 L1-R1 was utilized to detect individual or simultaneous presence of CPV1 and CPV2 nucleic acids in whole DNAs extracted from 18 previously isolated clinical samples (S), which were either singly or co-infected with CPV1 and/or CPV2. M: DNA marker; NC: negative control.

**Figure 4 ijms-26-04391-f004:**
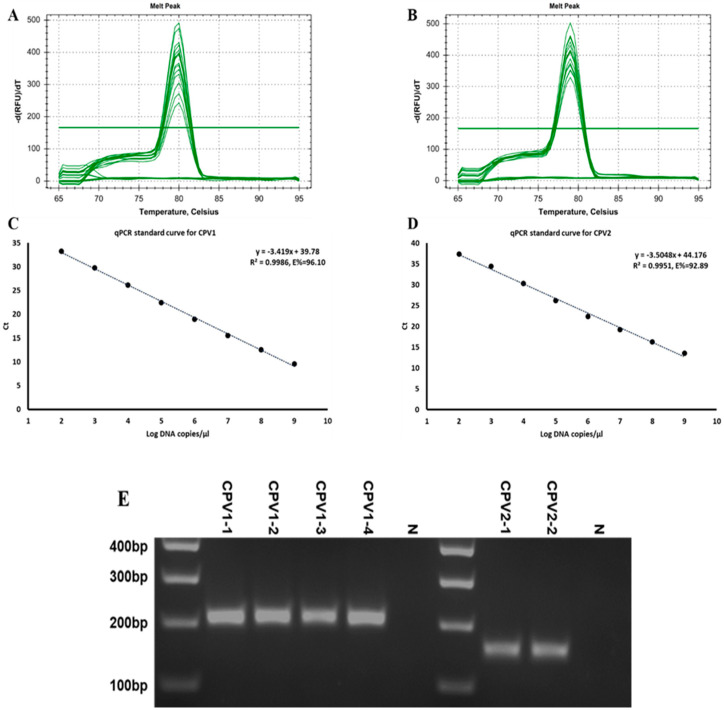
Relative quantification of CPV1 and CPV2 viral content by RT-qPCR. The primer pairs CPV1/2−L1-F0+CPV1−L1−R0 and CPV1/2−L1−F0+CPV2−L1−R0 were utilized to quantify the viral content in 4 CPV1 clinical samples and 2 CPV2 clinical samples, respectively. (**A**,**B**) Melt curves from RT−qPCR amplification confirmed the absence of primer–dimer artifacts and the specificity of the reaction. (**C**,**D**) Standard curves for the CPV1−L1 and CPV2−L1 genes, demonstrating the efficiency and linearity of the RT−qPCR assays, respectively. (**E**) Gel electrophoresis of RT−qPCR reaction products from clinic samples. M: DNA marker; N: negative control.

**Figure 5 ijms-26-04391-f005:**
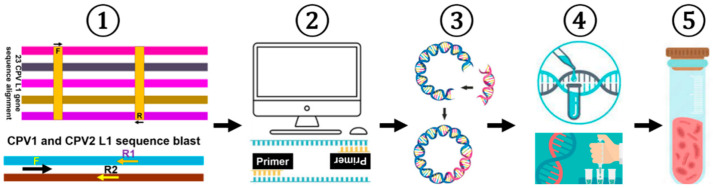
A schematic diagram outlining the experimental workflow for CPV detection and quantification. First, the L1 sequences of 23 CPV genotypes were aligned as described in Materials and Methods, while a focused blast of CPV1 and CPV2 sequences were conducted. Second, based on the analyses, a series of primers was designed, including type-specific primers for amplifying the full-length L1 gene of CPV1 and CPV2, general primers for detecting all CPVs, and type-specific primers for quantifying CPV1 and CPV2. Third, the full-length L1 genes of CPV1 and CPV2 were separately cloned into the pEF-GFP vector, while partial L1 fragments (~700 bp) from other CPV types were synthesized and inserted into the pUC57 vector, respectively, thereby generating standard or positive plasmids. Fourth, PCR and qPCR detection methods were optimized for improved CPV detection and quantification using these standard or positive plasmids. Fifth and finally, the developed assays were applied to clinical samples, followed by a comparison with a previously reported approach to evaluate its accuracy and efficiency.

**Table 1 ijms-26-04391-t001:** Primers designed and used in this study.

Primer Category	Primers	Sequence (5′-3′)	Amplicon Size	Application
General detection primers	CPV-General23-F0	TtCCHGAcCCtAAcaaDTTTGC	673~704 bp	General detection CPV1-23
	CPV-General23-R0	TCacTHGaVacKAGaGABCCACT		
CPV1/CPV2 typing and identification primers (triple primer method)	CPV1/2L1-F	ATAAGTCAGGAGTTCCAATTGACATAG	300/400 bp or 350/450 bp (depending on the downstream primers)	Simultaneously detect and distinguish between CPV1 and CPV2 infections
	CPV1L1-R1	TCTTGTTGCCCTATGTTTATAGTTAAG	400 bp (in combination with CPV1/2L1-F)	
	CPV1L1-R2	TGAGCTTGCCTAATCCAATATGGTC	300 bp (in combination with CPV1/2L1-F)	
	CPV2L1-R1	TGAGGTAGGTTCTGTATGATGAAGG	450 bp (in combination with CPV1/2L1-F)	
	CPV2L1-R2	CGACCGTTACAAACAGGTTGTTGTTC	350 bp (in combination with CPV1/2L1-F)	
Full-length L1 gene amplification primers	CPV1L1-U	ATGGCGGTTT GGCTTCCTGC ACAG	1512 bp	Amplify full-length CPV1-L1 gene
	CPV1L1-D	TTATTTGCGTTTGCGTTTCACAGC		
	CPV2L1-U	ATGGCCTTTT GGATCCAATCTCAAG		Amplify full-length CPV2-L1 gene
	CPV2L1-D	TTATGCCTTCCTTCTTTTCTTTGAAG		
Quantitative detection primers (for qPCR)	CPV1/2-L1-F0	TCAGGAGTTCCAATTGACATAG	140/190 bp (depending on downstream primers)	Specific quantification of CPV1 or CPV2 viral content
	CPV1-L1-R0	CTCTTGTTGCATAGGTATCTTTG	190 bp (in combination with CPV1/2-L1-F0)	
	CPV2-L1-R0	CCCTTACAAACATGTGTCTGC	140 bp (in combination with CPV1/2-L1-F0)	

**Table 2 ijms-26-04391-t002:** Relative quantification of CPV1 and CPV2 viral content in clinical samples using real-time PCR.

Genotypes	Samples	Ct Value (Mean ± SD, n = 3)	Viral Content (Copies/µL)
CPV1	1	11.72 ± 0.14	1.1 × 10^8.21±0.04^
	2	17.97 ± 0.23	1.1 × 10^6.38±0.07^
	3	10.43 ± 0.08	1.1 × 10^8.59±0.02^
	4	12.50 ± 0.12	1.1 × 10^7.98±0.04^
CPV2	1	13.87 ± 0.13	1.1 × 10^8.65±0.04^
	2	14.97 ± 0.05	1.1 × 10^8.33±0.01^

**Table 3 ijms-26-04391-t003:** Analysis of CPV epidemiology using suspected clinical samples collected from various animal hospitals across the United States, 2017–2021.

Year	CPV1 (n)/ Percentage (%)	CPV2 (n) /Percentage (%)	Other Genotypes (n) /Percentage (%)	Co-Infection by CPV1 and CPV2 (n)/Percentage (%)	Annual Total (n)
2017	17 (80.95%)	3 (14.29%)	1 (4.76%)	0 (0.00%)	21
2018	32 (94.12%)	0 (0.00%)	1 (2.94%)	1 (2.94%)	34
2019	19 (63.33%)	5 (16.67%)	1 (3.33%)	5 (16.67%)	30
2020	18 (75.00%)	2 (8.33%)	2 (8.33%)	2 (8.33%)	24
2021	23 (85.19%)	1 (3.70%)	1 (3.70%)	2 (7.40%)	27
Total (n)/ Percentage	109 (80.15%)	11 (8.09%)	6 (4.41%)	10 (7.35%)	136

**Table 4 ijms-26-04391-t004:** Epidemiological analysis of CPV epidemiology based on samples collected from animal hospitals, rescue centers and /breeding facilities across China.

Sample Source	Sample Type	CPV1 (n)/ Percentage (%)	CPV2 (n) /Percentage (%)	CPV1+CPV2 (n) /Percentage (%)	Total (n)
Animal hospital	Oral swab	0 (0.00%)	4 (19.48%)	2 (9.52%)	21
Animal Rescue center	Oral swab	32 (62.74%)	14 (27.45%)	5 (9.80%)	51
Professional dog breeding facility	Oral swab	6 (33.33%)	10 (55.56%)	2 (11.11%)	18

## Data Availability

Data is available upon request.

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
