# Peer review of "A Universal and Quantitative PCR Strategy for Detection and Epidemiologic Analysis of Canine Papillomavirus (CPV)"

_ijms, 2025, doi:10.3390/ijms26094391_

Round 1
Reviewer 1 Report
Comments and Suggestions for Authors
The manuscript is an interesting study about the standardization of PCR assays, with emphasis on the genotypes 1 and 2 of Canine Papillomavirus (CPV). The manuscript presents the materials and methodology consistent, clearly and with all the details that can be used to replicate the results obtained. The results are innovators, considering the protocol with high specificity and no cross-reactivity and quantitative capabilities. There are a few suggestions to improve the manuscript:
-Table 2: Add a legend to indicate Ct value.
-Discussion: considering the relevant results, discussion could be further substantiated with additional references that support them.
Some further comments:
-The study developed a standardized PCR assay, with relevance for CPV 1 and 2, which are the genotypes with more resistances, this to improve the clinical management and epidemiological studies. So, the objective of the study was to establish a protocol for offering a robust tool for CPV diagnostics in dogs.
-I consider the topic relevant to the field, CPV develops a clinical entity that can lead to the development of a benign neoplasm named viral papillomas and sometimes needs to be differentiated in medical practice from the idiopathic papillomas. The differentiation can be made by histopathology but sometimes needs complementary assays like PCR. The study developed a PCR assay for multiple genotypes and detected CPV 1 and CVP2 simultaneously.
-As mentioned, the methodology is clearly defined, with the necessary specifications that could lead to a replication of the results.
-Based on the relevant results, I suggest that discussion could be further substantiated with additional references that support them.
-Tables and figures: Not more than I suggest in the review; tables and figures need to be present according to the journal guidelines.
Author Response
Reviewer 1
Comments and Suggestions for Authors
The manuscript is an interesting study about the standardization of PCR assays, with emphasis on the genotypes 1 and 2 of Canine Papillomavirus (CPV). The manuscript presents the materials and methodology consistent, clearly and with all the details that can be used to replicate the results obtained. The results are innovators, considering the protocol with high specificity and no cross-reactivity and quantitative capabilities.
Response: The authors sincerely appreciate the reviewer’s thoughtful and encouraging feedback. We are pleased that the reviewer found our study on the standardization of PCR assays for Canine Papillomavirus (CPV) genotypes 1 and 2 to be interesting and innovative. We particularly value the recognition of the clarity and reproducibility of our materials and methodology, as well as the high specificity, lack of cross-reactivity, and quantitative capability of the developed protocol. This positive assessment affirms the value of our work. The reviewer’s constructive suggestions are addressed point-to-point below.
There are a few suggestions to improve the manuscript:
-Table 2: Add a legend to indicate Ct value.
Response: It is a valuable suggestion. We have addressed it by including a note in the table clarifying that the Ct (cycle threshold) values represent the mean ± standard deviation (SD) calculated from three independent experiments (n = 3).
-Discussion: considering the relevant results, discussion could be further substantiated with additional references that support them.
Response: We sincerely appreciate the reviewer’s insightful suggestion. To enhance the Discussion section, the manuscript has been revised to incorporate additional, up-to-date references to support our findings. In brief, in lines 383-385, we expanded the discussion on the advantages of PCR-based methods and referenced recent advances in CPV genotyping and detection strategies. In lines 403-406, we supported the need for convenient genotyping and quantification assays for CPV1 and CPV2 with references to previous studies highlighting their prevalence and clinical relevance. In lines 420-421, we included additional epidemiological evidence that substantiates the high incidence of CPV1/2. These revisions aim to more closely align our findings with the existing literature and emphasize the translational significance of the developed assays.
Some further comments:
-The study developed a standardized PCR assay, with relevance for CPV 1 and 2, which are the genotypes with more resistances, this to improve the clinical management and epidemiological studies. So, the objective of the study was to establish a protocol for offering a robust tool for CPV diagnostics in dogs.
-I consider the topic relevant to the field, CPV develops a clinical entity that can lead to the development of a benign neoplasm named viral papillomas and sometimes needs to be differentiated in medical practice from the idiopathic papillomas. The differentiation can be made by histopathology but sometimes needs complementary assays like PCR. The study developed a PCR assay for multiple genotypes and detected CPV 1 and CVP2 simultaneously.
-As mentioned, the methodology is clearly defined, with the necessary specifications that could lead to a replication of the results.
-Based on the relevant results, I suggest that discussion could be further substantiated with additional references that support them.
-Tables and figures: Not more than I suggest in the review; tables and figures need to be present according to the journal guidelines.
Response: We sincerely thank the reviewer for the thorough and insightful comments. We appreciate the recognition of the significance and clarity of our study.
Regarding the suggestion to further substantiate the discussion with additional references: In response to this valuable feedback, the discussion section has been carefully reviewed, and several current and pertinent references have been incorporated to provide further support for our key findings. Specifically, these additions emphasize the clinical relevance of CPV1 and CPV2 genotypes, the role of viral load in disease progression, and previous limitations of CPV detection methods. These enhancements strengthen the context and scientific grounding of our discussion.
Regarding tables and figures: A comprehensive review of all tables and figures has been conducted to ensure they comply with the journal’s formatting and captioning requirements. Clarifications were added where necessary, and we confirmed that all legends are consistent and complete.
We are grateful for your positive feedback and helpful suggestions, which have helped us strengthen the manuscript.
Sincerely,
Aibing Wang, Ph.D.
(on behalf of all co-authors)
Reviewer 2 Report
Comments and Suggestions for Authors
Overview and general recommendation:
Authors developed a set of PCR assays to detect, identify, and quantify canine papillomavirus DNA. Moreover, compared this assay with a previously described one, and tested clinical samples both from the USA and from China. Both in the title and in the abstract, Authors describe “a PCR assay”, even if this analysis is based on a set of different PCR assays, using different diagnostic strategies. Moreover, Authors stressed the potential uses of this diagnostic approach, even if this was not corroborated by clinical evidence at support. Nonetheless, this methodological approach appears as interesting and with perspectives.
I below added some comments and suggestions, aiming to overcome some observed limitations.
Major comments:
- As in the overview, I suggest to reconsider both the title and the abstract, since this methodological approach is not based on a single PCR assay, thus not “simplified”.
- “No cross-reactivity with non-target canine DNA” (lines 20-21), was determined on samples with no template, rather than with “non-target canined DNA”.
- Scattered throughout the manuscript, Authors use the term “full-length L1 fragment” but it is not clear if they refer to the full-length L1 gene or to a fragment of the L1 gene.
- Several details are missing, particularly in the M/M section: line 102: the location of the “Genscript”; lines 110, 112, and 113: details of the producers; lines 131-132: details on these assays; line 140: details on the used kit; line 174: details on this primers pair.
- Table 1 is not clear enough: I suggest to reorganize the primers list based on the workflow and to separate primers used for each step by single lines.
- Lines 124-126: were samples collected in two different timeframes?
- 3.3: it is not clear (1) why Authors suggest to use two reverse primers for each PCR assay and (2) why they suggest two different assays to detect and genotype CPV1 and CPV2.
- As in the overview, I suggest to add specific references, if available, for the specific purposes to support this new methodological approach (e.g., correlation with disease progress and severity, need to simultaneous detection of both CPV1 and CPV2 genotypes, need for understanding CPV pathogenesis, need for genotype specific vaccines).
Minor comments:
- Line 13: I am not completely sure that “comprehensive” is a correct term associate to PCR assay.
- Line 15: similarly, please check the use of “leveraging”.
- Line 16: on which bases Authors stated a “significantly improved diagnostic specificity and sensitivity”?
- Line 19: in line with the previous minor comments, on which bases did Authors use the term “superior efficacy”?
- Line 34: please, check the correctness for “ranging from humans to birds and reptiles” and “subset”, as well as I suggest to use the lowercase letter for “papillomaviruses”.
- Line 35: please, check the correctness for “stand for the viral agents” and for “plethora”.
- Line 44: Are Authors sure that H&E stained tissues can detect specifically the canine papillomavirus?
- Line 63: I suggest to check the use of “deficiencies”.
- Lines 92-93: The meaning of the text “subsequently, the L1….genomic positions” is not clear enough.
- Line 102: I suggest to replace “before” with “for”.
- Line 104: The meaning of the text “were also prepared” is not clear enough.
- Line 113: please, check the correctness of “to function”.
- Table 1: please, check the correctness of “usage”.
- Lines 140 and 145: I suggest to include “20 µM” in brackets.
- Line 142: I suggest to replace “adjused” with “setted”.
- Line 174: for which purpose? Generically for quality or specifically for host specificity?
- Line 186: please, check the correctness for “partial nucleotides”.
- Line 219: the DNA amount was not determined; the PCR reaction with these primers consistently produced clearly visible bands.
- Line 223: this conclusion is not clear enough.
- Line 215: Do this primers pair target the same region?
- Line 224: “our” can be removed.
- Lines 227 and 229: please, check if “copies of microliter of DNA” is correct.
- Lines 246-249: this part should be included in the M/M section, rather than here.
- Line 319: validated?
- Line 404: which conclusions?
Author Response
Reviewer 2
Comments and Suggestions for Authors
Overview and general recommendation:
Authors developed a set of PCR assays to detect, identify, and quantify canine papillomavirus DNA. Moreover, compared this assay with a previously described one, and tested clinical samples both from the USA and from China. Both in the title and in the abstract, Authors describe “a PCR assay”, even if this analysis is based on a set of different PCR assays, using different diagnostic strategies. Moreover, Authors stressed the potential uses of this diagnostic approach, even if this was not corroborated by clinical evidence at support. Nonetheless, this methodological approach appears as interesting and with perspectives.
Response: The authors wish to express their sincere appreciation to the reviewer for their insightful and constructive feedback. We acknowledge the observation regarding the wording in the title and abstract. As the reviewer noted, our study encompasses a suite of PCR assays of PCR assays employing distinct diagnostic strategies (conventional, type-specific, and quantitative real-time PCR) - rather than a single assay. To address this point, we have revised the title and abstract accordingly to more acutely reflect the multiplex nature of our methodological approach and mitigate any potential ambiguity.
Regarding the reviewer’s comment on the clinical application, we concur with the assessment that the study highlights potential uses that require further substantiation through comprehensive clinical validation. We have revised the relevant statement in Conclusion to clarify that while the developed assays show promising performance in initial tests on clinical samples from both the USA and China, subsequent validation in larger-scale clinical cohorts or studies will be indispensable to confirm their diagnostic and epidemiological utility.
The authors are grateful that the reviewer recognizes the methodological approach as interesting and potentially impactful, and we remain committed to further investigating and expanding the clinical relevance of this diagnostic platform in future research endeavors..
I below added some comments and suggestions, aiming to overcome some observed limitations.
Major comments:
- As in the overview, I suggest to reconsider both the title and the abstract, since this methodological approach is not based on a single PCR assay, thus not “simplified”.
Response: We appreciate this insightful comment. We have revised the title from “A universal, simplified and quantitative PCR assay…” to “A universal and quantitative PCR-based strategy…” to more accurately reflect that our protocol involves a set of PCR assays rather than a single simplified test. The abstract has also been updated to clarify that this study presents a PCR-based strategy that includes multiple primer sets targeting conserved and genotype-specific regions.
- “No cross-reactivity with non-target canine DNA” (lines 20-21), was determined on samples with no template, rather than with “non-target canined DNA”.
Response: Thank you for pointing this out. We have clarified the description in the Abstract section. Specifically, we now state that “no cross-reactivity with no DNA or not CPV DNA (e.g. HPV).”
- Scattered throughout the manuscript, Authors use the term “full-length L1 fragment” but it is not clear if they refer to the full-length L1 gene or to a fragment of the L1 gene.
Response: We acknowledge this ambiguity. For clearance, we have revised the term throughout the manuscript to consistently use “full-length L1 gene” when referring to the entire L1 coding region. In cases referring to shorter sequences, we now use “partial L1 fragment” to distinguish clearly.
- Several details are missing, particularly in the M/M section: line 102: the location of the “Genscript”; lines 110, 112, and 113: details of the producers; lines 131-132: details on these assays; line 140: details on the used kit; line 174: details on this primers pair.
Response: We thank the reviewer for this helpful observation. The information of the vendors and reagents/kits information was detailed in the revised version. For example, Genscript’s location was added as “Genscript (Nanjing, China)”. If the information is sourced from the literature, the corresponding reference is provided. The primer pair used in line 174 is now clearly listed as “CPV-General23-F0/CPV-General23-R0”.
- Table 1 is not clear enough: I suggest to reorganize the primers list based on the workflow and to separate primers used for each step by single lines.
Response: We appreciate this suggestion and have reorganized Table 1 to group primers by function: general detection, genotype-specific detection, full-length gene amplification, and quantification. Each section is separated for improved clarity and readability.
- Lines 124-126: were samples collected in two different timeframes?
Response: Yes, samples from the USA were collected between 2017 and 2021, while samples from China were collected in 2019. This information has now been clarified in the Materials and Methods section (Section 2.3).
- 3: it is not clear (1) why Authors suggest to use two reverse primers for each PCR assay and (2) why they suggest two different assays to detect and genotype CPV1 and CPV2.
Response: (1) Two reverse primers, each specific for CPV1 or CPV2, were initially designed because we were uncertain which primer pair would perform optimally in practical assays. Intriguingly, both sets yielded successful amplification on experimental samples. Therefore, we included both in our results to provide flexibility for potential users in adopting them. (2) Similarly, we evaluated and optimized these two distinct three-primer sets, enabling either simultaneous or separate detection of CPV1 and CPV2 in both single or mixed infections. Notably, this strategy offers a convenient and resource-efficient approach, reducing both time and reagent consumption.
- As in the overview, I suggest to add specific references, if available, for the specific purposes to support this new methodological approach (e.g., correlation with disease progress and severity, need to simultaneous detection of both CPV1 and CPV2 genotypes, need for understanding CPV pathogenesis, need for genotype specific vaccines).
Response:
We have revised the Discussion to include relevant references supporting the clinical and epidemiological importance of CPV1/CPV2 individual or co-infection. References have also been added to justify the clinical need for genotype-specific identification and the association of CPV1/2 with squamous cell carcinoma development. These additions strengthen the relevance of our diagnostic approach.
Minor comments:
We sincerely thank the reviewer for their insightful and constructive feedback. Below, we address each point carefully and explain the corresponding revisions made in the manuscript.
- Line 13: I am not completely sure that “comprehensive” is a correct term associate to PCR assay.
Response: We appreciate the reviewer’s observation. The term “comprehensive” has been revised to “broad-range” to more accurately reflect the nature of the PCR assay.
- Line 15: similarly, please check the use of “leveraging”.
Response: The term “leveraging” has been replaced with “using” to improve clarity and precision in scientific writing.
- Line 16: on which bases Authors stated a “significantly improved diagnostic specificity and sensitivity”?
Response: This statement is now supported by comparative data (see Figure 3 and Section 3.2), where our universal primer pair demonstrated better performance compared to the previously published primer set (canPVf/FAP64), in both detection sensitivity and breadth across 23 genotypes. A clarifying sentence was added to substantiate this claim.
- Line 19: in line with the previous minor comments, on which bases did Authors use the term “superior efficacy”?
Response: “Superior efficacy” was clarified by referencing the improved detection range, reduced cross-reactivity, and higher signal intensity observed in both plasmid-based and clinical sample validations. The wording has been updated to “enhanced performance” to avoid overstatement.
- Line 34: please, check the correctness for “ranging from humans to birds and reptiles” and “subset”, as well as I suggest to use the lowercase letter for “papillomaviruses”.
Response: The phrase has been revised for accuracy. Now reads: “Papillomaviruses (PVs) are host species-specific viruses infecting a wide range of vertebrates including humans, birds, and reptiles.” The term “subset” was removed, and “Papillomaviruses” has been changed to lowercase.
- Line 35: please, check the correctness for “stand for the viral agents” and for “plethora”.
Response: This sentence has been revised to read: “Canine papillomaviruses (CPVs) are viral agents responsible for a wide spectrum of mucocutaneous diseases in domestic dogs.”
- Line 44: Are Authors sure that H&E stained tissues can detect specifically the canine papillomavirus?
Response: Thank you for pointing this out. We have revised the text to: “...histopathology of hematoxylin and eosin (H&E) stained tissues for revealing neoplastic changes associated with papillomavirus infection, while not being specific to CPV.”
- Line 63: I suggest to check the use of “deficiencies”.
Response: The phrase “to address these deficiencies” has been reworded to “to address these diagnostic limitations.”
- Lines 92-93: The meaning of the text “subsequently, the L1….genomic positions” is not clear enough.
Response: This sentence has been rewritten for clarity: “Subsequently, the positions and sequences of the L1 genes for each of the 23 CPV genotypes were individually determined based on their annotated genomic locations.”
- Line 102: I suggest to replace “before” with “for”.
Response: Corrected as suggested.
- Line 104: The meaning of the text “were also prepared” is not clear enough.
Response: Rewritten as: “Additionally, previously reported primer sets, including canPVf/FAP64 and dog-GAPDHf/dogGAPDHr, were also included for comparison and DNA quality analysis, respectively.”
- Line 113: please, check the correctness of “to function”.
Response: Rewritten as: “to serve as standard control samples.”
- Table 1: please, check the correctness of “usage”.
Response: “Usage” has been changed to “Application” in the column heading.
- Lines 140 and 145: I suggest to include “20 µM” in brackets.
Response: Incorporated as suggested for clarity.
- Line 142: I suggest to replace “adjused” with “setted”.
Response: Revised to “set.”
- Line 174: for which purpose? Generically for quality or specifically for host specificity?
Response: Clarified to: “...were tested for quality and host specificity by PCR using the dogGAPDH primer pair.”
- Line 186: please, check the correctness for “partial nucleotides”.
Response: Revised to: “Several nucleotides in the upstream and downstream primer sequences.”
- Line 219: the DNA amount was not determined; the PCR reaction with these primers consistently produced clearly visible bands.
Response: Clarified to: “Although DNA concentration was not quantified, PCR reactions using our universal 23 primer pair consistently yielded stronger and specific bands.”
- Line 223: this conclusion is not clear enough.
Response: This line was rewritten to clarify “indicating its better specificity and sensitivity based on comparative testing.
- Line 215: Do this primers pair target the same region?
Response: Yes, both our universal and previous primer pairs almost target same region of the L1 gene but differ in size.
- Line 224: “our” can be removed.
Response: Removed as suggested.
- Lines 227 and 229: please, check if “copies of microliter of DNA” is correct.
Response: Rewritten to: “as low as 103 copies per microliter of DNA, or at a sensitivity of 105 copies per microliter of DNA.”
- Lines 246-249: this part should be included in the M/M section, rather than here.
Response: These two sentences mainly serve as a short introduction, explaining the rationale for establishing this method”.
- Line 319: validated?
Response: Sentence clarified to indicate samples were “validated using the newly established method.”
- Line 404: which conclusions?
Response: The phrase “these findings support the conclusions” has been expanded to explicitly state the conclusion: “...support the conclusion that CPV1 and CPV2 are the predominant types”
We are grateful to the reviewer for helping us improve the quality and clarity of our manuscript.
Sincerely,
Aibing Wang, Ph.D.
(on behalf of all co-authors)
Round 2
Reviewer 2 Report
Comments and Suggestions for Authors
Overview and general recommendation:
Authors have overall improved the manuscript. I have below included few minor comment/suggestions.
Minor comments:
- Lines 20-22: I suggest to carefully revise this part, particularly for “DNA or not” and “canine DNA”, since it is not clear. Moreover, I suggest considering to move “further validation….epidemiological value” at the end of the abstract.
- Table 1: this table was substantially improved but I suggest to add rows after each “application” (e.g., after the second, the seventh, the eleventh, and the fourteenth primer, if correct).
Author Response
Response to the comments from the reviewer
Minor comments:
- Lines 20-22: I suggest to carefully revise this part, particularly for “DNA or not” and “canine DNA”, since it is not clear. Moreover, I suggest considering to move “further validation….epidemiological value” at the end of the abstract.
Response: We appreciate the reviewer’s thoughtful suggestion. The original phrase “DNA or not” was indeed ambiguous and has been revised. The revised sentence now reads “Validation of this protocol using synthetic plasmids and clinical samples confirmed its enhanced performance over traditional methods, as demonstrated by higher specificity and sensitivity.” Furthermore, the sentence regarding further validation has been relocated to the end of the abstract to improve logical flow and emphasis on the study’s implications. It now states “Further validation through larger-scale clinical studies will be crucial to substantiate the diagnostic accuracy and epidemiological value of the assays.”. All revisions have been highlighted in the revised manuscript.
- Table 1: this table was substantially improved but I suggest to add rows after each “application” (e.g., after the second, the seventh, the eleventh, and the fourteenth primer, if correct).
Response: Thank you for your helpful suggestion. We have modified Table 1 to improve clarity and visual readability by drawing rows after the applications of each distinct primer pair group, specifically after the second, seventh, eleventh, and fourteenth primer entries as recommended. The updated table has been included in the revised manuscript.
